# The Cause–Effect Model of Master Sex Determination Gene Acquisition and the Evolution of Sex Chromosomes

**DOI:** 10.3390/ijms26073282

**Published:** 2025-04-01

**Authors:** Zhanjiang Liu, Dongya Gao

**Affiliations:** Department of Biology, College of Arts and Sciences, Tennessee Technological University, Cookeville, TN 38505, USA

**Keywords:** sex determination, sex differentiation, master sex determination gene, recombination suppression, chromosome inversion, sex chromosome evolution

## Abstract

The canonical model of vertebrate sex chromosome evolution predicts a one-way trend toward degradation. However, most sex chromosomes in lower vertebrates are homomorphic. Recent progress in studies of sex determination has resulted in the discovery of more than 30 master sex determination (MSD) genes, most of which are from teleost fish. An analysis of MSD gene acquisition, recombination suppression, and sex chromosome-specific sequences revealed correlations in the modes of MSD gene acquisition and the evolution of sex chromosomes. Sex chromosomes remain homomorphic with MSD genes acquired by simple mutations, gene duplications, allelic variations, or neofunctionalization; in contrast, they become heteromorphic with MSD genes acquired by chromosomal inversion, fusion, and fission. There is no recombination suppression with sex chromosomes carrying MSD genes gained through simple mutations. In contrast, there is extensive recombination suppression with sex chromosomes carrying MSD genes gained through chromosome inversion. There is limited recombination suppression with sex chromosomes carrying MSD genes gained through transposition or translocation. We propose a cause–effect model that predicts sex chromosome evolution as a consequence of the acquisition modes of MSD genes, which explains the evolution of sex chromosomes in various vertebrates. A key factor determining the trend of sex chromosome evolution is whether non-homologous regions are created during the acquisition of MSD genes. Chromosome inversion creates inversely homologous but directly non-homologous sequences, which lead to recombination suppression but retain recombination potential. Over time, recurrent recombination in the inverted regions leads to the formation of strata and may cause the degradation of sex chromosomes. Depending on the nature of deletions in the inverted regions, sex chromosomes may evolve with dosage compensation, or the selective retention of haplo-insufficient genes may be used as an alternative strategy.

## 1. Introduction

The mechanism of sex determination in vertebrates is enormously diverse, especially in teleost fish [1,2], ranging from unisexuals, hermaphroditism, and environmental sex determination to genetic sex determination [3]. With genetic sex determination, 30 distinct MSD genes have been identified from vertebrates under various sex determination (SD) systems, including XY, ZW, and multiple sex chromosomes [1,4]. Despite such diversity, the molecular pathways and downstream players are generally conserved (Table 1). The basic operation of the sex determination network occurs through the dynamics of the opposing male and female pathways to ensure a 1:1 male/female ratio [5], thus requiring the SD gene to be expressed in a temporally, spatially, dose-, and temperature-sensitive fashion. The gene products in the male pathway and female pathway are antagonistic, and their expression is negatively regulated against each other (Figure 1).

As with the MSD genes, sex chromosomes in vertebrates have evolved at differing paces and to various degrees, ranging from entirely homomorphic in many species of lower vertebrates to highly degraded and heteromorphic in higher vertebrates. The canonical model of sex chromosome evolution predicts a one-way trend toward the degeneration and degradation of the sex chromosome carrying the MSD gene [6,7,8,9]. However, empirical evidence in support of this theory does not go beyond mammals and birds. The sex chromosomes in reptiles, amphibians, and teleost fish can be homomorphic or heteromorphic, and most of them are homomorphic. A popular explanation is that the sex chromosomes in lower vertebrates are young, but both homomorphic and heteromorphic sex chromosomes are present across all lower vertebrates, ranging from cartilaginous fish, teleost fish, and amphibians to reptiles; some homomorphic sex chromosomes are older than some highly differentiated heteromorphic sex chromosomes, directly challenging the canonical theory of sex chromosome evolution. The availability of genome sequences and known MSD genes from various vertebrate species, especially from lower vertebrates, has made it possible to determine: (1) how MSD genes were acquired; (2) how sex chromosomes evolve in relation to the modes of MSD gene acquisition; and (3) how MSD genes affect the evolution of sex determination systems. Here, we present a cause–effect model that states that the mode of MSD acquisition determines the evolution of the sex chromosomes. This model explains the evolution of vertebrate sex chromosomes in relation to the MSD genes and their chromosome karyotypes.

**Figure 1 ijms-26-03282-f001:**
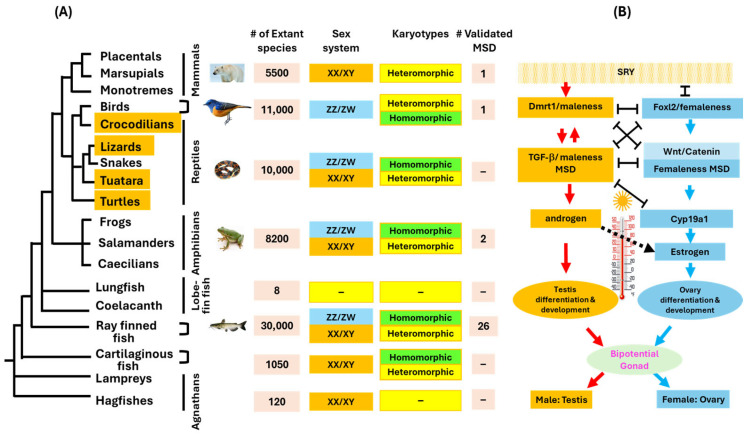
Schematic representation of sex determination systems, sex chromosomes and master sex determination (MSD) genes and their association with temperature (**A**). Note that the majority of homeothermic mammals have the XY sex determination system with just one validated MSD gene, Sry, although a multi-chromosome sex determination system has been found for monotremes, with their MSD genes (Amh) not conclusively determined. The majority of birds have heteromorphic sex chromosomes, but emu birds have homomorphic sex chromosomes [10]. The groups in which both genetic sex determination and temperature sex determination exist are colored in orange. (**B**) Male and female pathways of sex determination, starting with the master sex determination gene, e.g., Sry in mammals, dmrt1 in birds, and various other genes in lower vertebrates. The male and female pathways are antagonistic to each other, but this process is regulated by the temperature, which affects sex differentiation through the expression of aromatase. Note that the SRY transcription factor is upstream of all other MSD genes.

Only one MSD gene and one SD system have been confirmed for homeothermic mammals and birds, respectively, but multiple MSD genes and SD systems have evolved from poikilothermic vertebrates (Figure 1). In mammals, Sry is both necessary and sufficient for sex determination [11]. In birds, dmrt1 works in a dose-sensitive fashion, where two copies of dmrt1 result in a male, while one copy results in a female [12]. Apparently, a two-fold difference in dmrt1 expression is sufficient to reach the threshold for the sex phenotype in birds. In reptiles, the ZW system was believed to be the only genetic SD system for over 50 years, until the recent demonstration of the XY SD system in boa and python snakes [13]. Similarly, in amphibians, a female genome-specific DM-domain gene on the W chromosome (DM-W) in the African clawed frog, *Xenopus laevis*, was the only MSD gene known from 8470 amphibian species until the recent discovery of MSD gene *bot1l* in the European green toad, *Bufo viridis*, with an XY system [14].

As poikilotherms living in aquatic environments, teleost fish must respond to broad temperature variations in the SD process. Additionally, the fertilization of most teleost fish occurs externally. In correlation, 28 distinct MSD genes have evolved in teleost fish (Figure 1), including transcriptional factors, TGF-β cytokines, steroidogenesis genes, and many “newcomers”. All are downstream players in the SD pathways, as compared to the MSD gene Sry in mammals. While Sry functions as a decisive MSD gene, most other MSD genes are quantitative in regard to dose and temperature sensitivity.

## 2. Molecular Mechanisms for the Acquisition of MSD Genes

The first step in sex chromosome evolution is the emergence of MSD genes. To serve as a sex chromosome, one gene on one of the autosome pairs must initially gain new functions or new expression patterns in favor of the male or female pathway, thereby becoming a master switch, activating the genes for either male or female development. As shown in Figure 2, the specific mechanism of MSD gene acquisition determines the pathway of sex chromosome evolution. In the context of sex chromosome evolution, we classify the mechanisms of MSD gene acquisition into four categories. (1) Simple mutations, such as allelic variations (including base substitutions, neofunctionalization, and subfunctionalization), gene duplications, and small deletions or insertions [15]. This category accounts for the largest numbers of known MSD genes in vertebrates (Table 2). Sex chromosomes in this category are characterized by not harboring any non-homologous sequences and, therefore, they stay homomorphic. Examples of this category include missense mutations in the coding regions; mutations in the regulatory sequences, such as promoters, enhancers, silencers, and splicing junctions; and gene duplications. (2) Translocations or transpositions of DNA carrying the MSD gene or small-sized inversions. Sex chromosomes in this category carry a limited region of non-homologous sequences that do not recombine, but they stay homomorphic. Examples of this category include MSD gene DM-W in the African clawed frog, sdY in salmonids, and amhr2 among silurid catfishes. (3) Large chromosomal inversions. An MSD gene is activated due to the juxtaposition of the MSD gene to a new regulatory context, which leads to changes in expression profiles and/or temporal or spatial regulation in favor of the male or female pathway. Sex chromosomes in this category carry large non-homologous regions that are inverted between the sex chromosomes, which reduces crossover, but they retain their recombination potential upon the formation of the recombination loop [16]. Reduced recombination leads to sequence degeneration, but any recombination and recurrent recombination in the inverted region would lead to duplication in one and deletion in the other chromosome [17,18], and they most likely require double crossover to survive. As a result, sex chromosomes in this category become heteromorphic. (4) Chromosome fusion or fission. Chromosome fusion or fission leads to non-homologous regions of large sizes and often the multi-chromosome SD system.

## 3. The Diversity of Master Sex Determination Genes in Vertebrates

After the discovery of the SRY gene in humans [11], it took 12 years to find the second vertebrate MSD gene, dmY, from medaka [92]. However, recent advances in sequencing technologies have drastically accelerated the pace. To date, 30 distinct MSD genes have been identified from vertebrates (Table 2), with the vast majority of these being identified from teleost fish. Only two MSD genes have been identified from amphibians, DM-W from the African clawed frog and *bod1l* from the European green toad [14]. Although both XY and ZW SD systems have been found in reptiles, no MSD genes have yet been confirmed. However, several key sex-determining genes, such as dmrt1 and amh, have been implicated in reptiles, reflecting the diversity of the sex determination mechanisms in this group as well. Additionally, some reptiles, like the bearded dragon, exhibit interactions between genetic sex determination (GSD) and temperature-dependent sex determination (TSD), highlighting the evolutionary flexibility of reptilian sex determination systems.

In teleost fish, 28 distinct MSD genes have been identified, including transcriptional factors, TGF-β cytokines, genes involved in steroidogenesis, and many “newcomers” (Table 2). Of the transcriptional factors, dmrt1 is the most popular, although the sox family of transcription factors, including sox2, sox3, and sox7, also serves as MSD genes. In addition, the transcriptional factors FIGLA-like and ptf1a were identified to be MSD genes. Of these, the dmrt1 and sox genes were well known as the “usual suspects” [93], but FIGLA-like and ptf1a are newcomers. The FIGLA-like gene was reported to be the key sex regulator of LG1 sex determination in tilapia [32]; it encodes a protein of 99 amino acids, including a 45-amino-acid basic helix–loop–helix domain specifically expressed in the testis. While the autosomal FIGLA gene is a femaleness gene promoting ovary formation, the FIGLA-like gene on the Y chromosome interferes with the functions of the autosomal FIGLA gene, leading to testis development [32]. Similarly, a truncated form of pancreas transcriptional factor 1 alpha (Ptf1a), named ptf1aY, was identified as an MSD gene in the Chinese longsnout catfish [33]. Five TGF-β genes, namely amh, amhr2, gdf6, gsdf, and bmpr1b, have been identified as MSD genes in teleost fish. In approximately a dozen species, genes involved in steroidogenesis have been identified as MSD genes, including hsd17b1, cyp19a1, hsdl1, sult1st6y, and fshrY. Of these, the functions of hsd17b1 and cyp19a1 are well known. 17β-Hydroxysteroid dehydrogenase 1 oxidizes or reduces the C17 hydroxy/keto group of androgens and estrogens and, hence, regulates the potency of these sex steroids, while cyp19a1 is a temperature-sensitive aromatase that converts androgens into estrogens. Therefore, these are generally regarded as femaleness genes, and they serve as MSD genes mostly in ZW systems (Table 2). In addition to these three groups, over a dozen of newcomers have been identified as MSD genes (Table 2). These include sdY, cephx1Y, paics, banf2, RIN3, zkY, pfpdz1, hydin, and gipc1 (a pdz domain-containing gene). The pathways for sex determination in these genes are unknown, with the exception of sdY, which is a truncated form of interferon regulatory factor 9; sdY functions as a sex determination gene in salmonids as a dominant negative regulator through its interaction with FOXL2 [94].

## 4. Convergent and Divergent Evolution of MSD Genes

Despite the convergent evolution of MSD genes in mammals and birds, most known MSD genes from lower vertebrates evolve independently. The identification of 30 distinct MSD genes indicates unlimited options among MSD genes, and the options go far beyond the usual suspects, with a trend of moving from upstream “master” to downstream players. Limited information is available for amphibians and reptiles, but the MSD genes evolve mostly independently in lower vertebrates, with some local convergence. As shown in Figure 3, many orders involve more than one MSD gene, reflecting the divergent evolution of MSD genes. Even if a specific MSD gene is found in multiple orders, the molecular mechanisms of their acquisition are different (Table 2). Of all teleost orders, Cichliformes has the largest known number, with seven MSD genes, including transcription factors (banf2 and FIGLA), TGF-β members (amh, amhr2, gsdf), and the newcomer category of unknown pathways (paics and RIN3). This is followed by Beloniformes and Pleuronectiformes, each with six known MSD genes, and then by Perciformes and Siluriformes, each with five known MSD genes. Convergent evolution, mostly through shared ancestry [95], does exist locally, mostly within the orders. The most dramatic is the conservation of sdY as the MSD gene across the entire group of salmonids [82,83]. The common GTF-β factors could be shared by several genera or families, but none go beyond the scope of the order.

## 5. The Canonical Model of Sex Chromosome Evolution

The canonical model of sex chromosome evolution includes four consecutive phases: (1) first, an MSD gene acquires a sex-determining function by mutation from a pair of autosomes; (2) recombination suppression occurs between the sex chromosomes; (3) the degeneration of the sex chromosome occurs with the accumulation of deleterious mutations and TE in the non-recombinational regions of the sex chromosomes (Y or W); (4) deletions of the degenerated sex chromosome lead to the decay of the sex chromosome and evolution of dosage compensation [6,7,8,9]. The diversity and evolutionary lability of sex chromosomes across the tree of life [97] have challenged the canonical model. The vast majority of the known sex chromosomes in lower vertebrates are homomorphic; recombination suppression is involved only in a small fraction of species with known sex chromosomes. Recombination suppression does not always cause the degradation of sex chromosomes; sex chromosome degeneration and degradation have been demonstrated only in a small fraction of cases in lower vertebrates [98]. Sex-antagonistic genes have not been widely identified, and, in a small number of examples, the linkage of genes and their related phenotypes cannot be automatically interpreted as causation [8,99]. The remarkable turnover of sex chromosomes in many systems, especially in teleost fish, does not support the inevitable linearity of sex chromosome evolution [99,100,101]. Dosage compensation has been observed in some but not all species with heteromorphic sex chromosomes. Instead, in some teleost fish species, the selective retention of haplo-insufficient genes is apparently an alternative strategy to cope with the imbalanced expression of X- or Z-linked genes [27,46]. All these situations differ from those in humans [102] and birds [103] and thus motivates the proposal of new theories of sex chromosome evolution.

## 6. The Proposed Cause–Effect Model for the Evolution of Sex Chromosomes

Based on analysis of the mechanisms of MSD gene acquisition, we propose a cause–effect model of sex chromosome evolution, which is depicted in Figure 2. The ways in which sex chromosomes evolve depends on how the MSD genes were acquired. Sex chromosomes carrying MSD genes acquired from simple mutations such as allelic variations, neofunctionalization, and gene duplication stay homomorphic, without recombination suppression or degeneration. Sex chromosomes carrying MSD genes acquired from translocation and transposition carry a small non-homologous region between the sex chromosomes; therefore, they develop recombination suppression within the transposed segments, but they remain homomorphic. Sex chromosomes carrying MSD genes acquired from chromosomal inversion carry inverted regions between the sex chromosomes. Recombination is reduced in the inverted region, but a single crossover in the inverted region causes deletion or duplication, which may be lethal to the gametes; they thus require double crossover to survive (Figure 4). Over time, such sex chromosomes degenerate and decay, especially with large inversions. A key factor for the evolution of sex chromosomes is whether non-homologous sequences are created during the acquisition of the MSD genes. Sex chromosomes with MSD genes acquired through simple mutations do not create non-homologous regions. In contrast, sex chromosomes with MSD genes acquired through transposition, inversion, fusion, and fission all create non-homologous regions between the sex chromosome pairs, leading to recombination suppression. Key differences in this model from the canonical sex conflict model include the following: (1) sex chromosome evolution resolves the structural problems created during the acquisition of MSD genes, such as the presence of inverted regions on the sex chromosome pairs; (2) recombination suppression is the result of inter-chromosomal non-homologous regions between the sex chromosome pairs and not of the intrachromosomal linkage of antagonistic alleles; (3) in spite of the reduced recombination due to inversion, recurrent recombination in inverted regions over time leads to deletions on the sex chromosomes. Sequence degeneration and the accumulation of transposable elements over time result in evolutionary strata.

### 6.1. Homomorphic Sex Chromosomes Without Recombination Suppression

The cause–effect model of sex chromosome evolution predicts that sex chromosomes harboring MSD genes acquired from simple mutations are homomorphic, with no recombination suppression or sequence degeneration (Figure 2). These sex chromosomes became sex chromosomes because a gene that they carried became the MSD gene. Allelic and/or neofunctionalization/subfunctionalization accounts for the largest number of teleost fish species with known MSD genes to date. Teleost fish underwent a third round of whole-genome duplication [104], and the neofunctionalization and subfunctionalization of ohnolog genes is part of the rediploidization process [15,105]. Similarly, in many teleost species with known MSD genes, tandem gene duplications are involved (Table 2). Tandem duplications occur frequently in teleost fish species, especially with cytokines and chemokines [106].

One may argue that these sex chromosomes have not evolved considerably, simply because they are young. However, the sex chromosomes included in this category include taxa from cartilaginous fish, teleost fish, amphibians, and reptiles. Among teleost fish, a broad spectrum of teleost orders is involved, ranging from the base of the teleost order, such as Clupeiformes, to the most advanced order of Tetraodontiformes, covering approximately 300 million years of evolutionary time. It is unlikely that all of these sex chromosomes are at the very beginning of their evolution. In contrast, in various medaka, both homomorphic and heteromorphic chromosomes have been found, where allelic variations have been found with homomorphic sex chromosomes (e.g., in *O. latipes*), but chromosomal inversion has been found with heteromorphic sex chromosomes (e.g., *O. javanicus*), even though their evolutionary timelines are similar [21].

### 6.2. Homomorphic Sex Chromosomes with Limited Regions of Recombination Suppression

The cause–effect model of sex chromosome evolution predicts that sex chromosomes harboring MSD genes acquired with insertions, translocations, or transpositions may develop a sex chromosome-specific region (often referred to as MSY for a male-specific Y under an XY sex system) related to the insertion. In the inserted segments, recombination suppression may be present; sequence degeneration may occur, but the region of recombination suppression or the degenerated sequences may be limited to the insertional segments or slightly larger, while the bordering homologous sequences should have continued homologous recombination. As a result, the sex chromosome-specific region may carry fixed haplotypes, but the sex chromosomes stay homomorphic. As listed in Table 2 and illustrated in Figure 2, good examples of this category are sex chromosomes in species of the *Silurus*, *Pangasianodon*, and *Pangasius* genera in the order of Siluriformes. A common insertion was found to have occurred at the *Pangasidae* base, which carried the MSD gene amhr2 [51,52,53,54]. After acquiring the MSD gene, the sex chromosomes evolved to the extent that sequence degeneration and recombination suppression were observed within the insertions but were limited to the sex chromosome-specific insertion. It should be noted that, here, the MSD gene is ancient, estimated to have emerged over 100 million years ago [107], which directly contradicts the interpretation that the limited evolution of the sex chromosomes was because the evolutionary time was short. Similarly, the sdY gene in all salmonids has evolved over approximately 60 million years, but the sex chromosomes continue to be homomorphic [83].

### 6.3. Homomorphic Sex Chromosomes with Extended Recombination Suppression

The cause–effect model of sex chromosome evolution predicts that sex chromosomes involving large chromosomal inversions develop recombination suppression in the region of inversion. If the sex chromosomes are young, they are observed as homomorphic sex chromosomes. In several cases in teleost fish, large SDRs have been observed (Table 2). Interestingly, the levels of sequence degeneration within the SDRs are low in such cases. It is likely that such sex chromosomes are still young; it is also possible that such sex chromosomes stay homomorphic because of other mechanisms of regulation that slow down the evolution of the sex chromosomes (see below).

### 6.4. Heteromorphic Sex Chromosomes

In contrast to the situations of homomorphic sex chromosomes, heteromorphic sex chromosomes evolved from major structural changes in the sex chromosomes. Chromosome inversion is probably the most frequent cause of the evolution of heteromorphic sex chromosomes. With chromosomal inversion, a promoter or enhancer may be directly juxtaposed to a gene close to the inversional junction. If such a gene is a maleness or femaleness gene, the chromosomal inversion may have activated a gene as an MSD gene. In addition to the direct juxtaposition of regulatory elements to bordering genes near the inversion junction, changes in the chromatin structure and architecture may also have the capacity to enable the activation or deactivation of genes within or near the inversion segment. While the genomic and epigenomic regulation of the multi-dimensional chromatin architecture is not well understood, it is increasingly recognized that such an architecture and its spatial regulation is important for sex chromosome evolution [84]. If inversion is involved in the acquisition of the MSD gene, the “new” chromosome carrying the inversion initially still has the DNA content, but the major change is the creation of non-homologous sequences between the pair of sex chromosomes in the inverted region. As demonstrated in the ninespine stickleback, inversions may have a role in both the evolution of sex determination systems and the differentiation of sex chromosomes [108].

Accurate chromosome segregation during meiosis relies on homology between the maternal and paternal chromosomes. Yet, by definition, heterogametic sex involving heteromorphic sex chromosomes lacks a homologous partner [109]. The presence of chromosomal inversions reduces recombination; however, recombination does occur in inverted regions, which causes unequal crossover, leading to segmental duplications or deletions in gametes. Once the recombination is suppressed, over time, the sequences degenerate, TEs accumulate, and, if no regulation occurs, deleterious mutations accumulate. Eventually, upon any recombination or rearrangement, deletions occur, leading to the decay of the sex chromosomes. The sex chromosomes eventually become stable when large, inverted regions are eliminated or are repositioned within heterochromatin regions, such as those close to the centromere, where there is no recombination.

While such processes have been well documented in the canonical model of sex chromosome evolution, as with those in mammals and birds, the best-studied teleost fish under this category is perhaps the threespine stickleback (*Gasterosteus aculeatus*), with a large SDR of 17.5 Mb [46]. The MSD gene amhY is located in the oldest region of the stickleback Y chromosome, close to the original inversion junction (stratum one), adjacent to the pseudoautosomal region. The three evolutionary strata suggested additional inversions and rearrangements on the Y chromosome, and such inversions were confirmed by genetic mapping [110]. The Y chromosome is less than 26 million years old, its sequence is degenerated, and it has lost the majority of the genes that are present on chromosome X, retaining just 44.1% of these genes [46]. However, this gene loss may not have been random, as many haplo-insufficient genes were retained on the Y chromosome [46].

### 6.5. Multiple and Unequal Sex Chromosome SD Systems

A multiple-chromosome sex system has been found to operate in various vertebrates, including both lower and higher vertebrates. However, they are special cases of the XY sex chromosome. The breakage of either or both sex chromosomes is the basis of multi-chromosome sex determination. Among mammals, monotremes have an X5/Y5 system and thus a multi-chromosome sex system [111]. Multi-chromosome sex determination has also been found in cartilaginous fish and teleost fish. In teleost fish, a multiple-sex-chromosome SD system has been adopted in a sizable number of species. The actual number is likely to be larger, but 75 multiple-sex-chromosome systems with 60 estimated independent origins have been documented to date [112]. Multiple-sex-chromosome systems can be viewed as special cases of heteromorphic sex chromosomes. In this context, the cause–effect model of sex chromosome evolution predicts that sex chromosomes derived from chromosome fusion or fission would create heteromorphic sex chromosomes; the extent of recombination suppression depends on the size of the fusion or fission segments, as well as on the size of additional inversions. Here, we present one example to show that the cause–effect model also fits multiple-sex-chromosome systems. In the spotted knifejaw (*Oplegnathus punctatus*), the X1X2X1X2/X1X2Y multiple-chromosome SD system operates. Genome sequencing has revealed large genomic regions of recombination suppression, namely 29.3 Mb of X1 (from 0 to 29.30 Mb) and 17.58 Mb of X2 (from the centromere to 17.58 Mb), associated with sex, among which a large inversion on X1 and the centromere on X2, as well as the SD locus, accounted for the observed recombination suppression. Sequence degeneration and gene loss were also observed [113].

## 7. Dosage Compensation and Epigenetic Regulation of Sex Chromosome Evolution

Dosage compensation equalizes the levels of expression of X- or W-linked genes between the sexes. While complete dosage compensation is well documented for mammals [114,115], a lack of global dosage compensation has been reported for birds, monotremes, and reptiles [116,117,118,119]. In teleost fish, studies on the tongue sole and threespine stickleback have indicated a lack of dosage compensation but the selective retention of dosage-sensitive genes [27,120]. In contrast, complete dosage compensation was observed in the guppy species *Poecilia parae* and *P. picta* [121,122]. This appears to be conflicting but may represent alternative strategies to cope with the imbalanced expression of sex chromosomes in species with heteromorphic sex chromosomes. In cases where large deletions had occurred from the Y chromosome, such as in the guppies, complete dosage compensation was needed, and this was observed. This may be an effective way to balance the expression of the lost genes, especially those that are dosage-sensitive. Alternatively, with the selective retention of dosage-sensitive genes, there is no need to develop dosage compensation, as observed with less degenerated sex chromosomes in the threespine stickleback and tongue sole [99,100].

Epigenetic regulation is involved in growth, reproduction, disease resistance, and stress responses in various fish species [123]. Several isolated studies have suggested the epigenetic regulation of sex determination and sex chromosome evolution. In the channel catfish, an epigenetically marked locus of 8.9 Mb was well aligned with the SDR, where recombination was completely suppressed [86,124]. The X chromosome was hypermethylated, leading to the silencing of the X-borne hydin gene. In contrast, the Y chromosome was hypomethylated, and the Y-borne hydin gene was expressed, serving as an MSD gene [87,125]. Similarly, in the threespine stickleback, the sex chromosomes had the majority (65%) of the differentially methylated CpG sites (DMS) in the genome, with hypermethylation in females and hypomethylation in males. Most interestingly, the DMS were predominantly located in the SDR, especially in strata 2 and 1, where recombination was suppressed [126]. Similar work was also conducted in two guppies, where differential methylation was observed in the testis, with hypermethylation in females and hypomethylation in males. Again, the DMS were found mostly in the SDR, particularly in stratum 2 and stratum 1, where the MSD gene was located [127]. These examples demonstrate the differential methylation of the sex chromosomes, especially within the SDR. In addition to the regulation of gene expression, one possibility is that hypermethylation in the SDR could block recombination, thereby protecting the inverted region from being deleted, possibly allowing the selective retention of dosage-sensitive genes. The inhibition of recombination during meiosis by hypermethylation is well documented in plants [128,129]. If DNA methylation plays a similar role within the SDR, especially with chromosomal inversion, hypermethylation in the SDR would de facto slow down the degeneration of the sex chromosomes. It is possible that such a mechanism could be used to selectively retain haplo-insufficient genes. Epigenetic regulation is also involved in dosage compensation. In a recent study, a long non-coding RNA (lncRNA), termed MAYEX, regulated dosage compensation, which balanced the expression of sex chromosomes, in reptiles [130]

## 8. Conclusions and Perspectives

Recent advances in sequencing technologies have allowed the rapid discovery of the enormously diverse MSD genes in vertebrates and the mechanisms of their acquisition. A number of factors leading to structural changes in chromosomes result in the formation of sex chromosomes. These changes, such as chromosomal inversions, occur due to errors in homologous recombination, DNA breakage, and repair. Contributing factors include misaligned meiotic crossover, double-strand breaks (caused by radiation, chemicals, or oxidative stress), transposable elements, and non-homologous end joining. Additionally, replication errors, chromosomal instability disorder, viral integration, and unequal sister chromatid exchange can also play a role. These mechanisms lead to either paracentric inversions (not involving the centromere) or pericentric inversions (involving the centromere), ultimately influencing gene expression, evolution, and genetic disorder. Multiple-sex-chromosome systems are a result of chromosomal breakage, often at the most repetitive regions of the chromosome. It is apparent that the trajectories of sex chromosome evolution are a result of the acquisition modes of MSD genes: sex chromosomes with MSD genes acquired through simple mutations do not involve recombination suppression or degeneration; in contrast, sex chromosomes with MSD genes acquired through chromosomal inversions involve recombination suppression, which leads to the accumulation of transposable elements and sequence degeneration. Moreover, when the inversions are large in size, they may lead to the decay of the sex chromosomes. A key factor determining the evolution of sex chromosomes is whether inverted regions are created between the sex chromosome pairs for the acquisition of MSD genes. The creation of long inverted regions between the sex chromosome pairs provides a driving force for sex chromosome evolution [18]. Depending on the sizes of the inversions and the nature of the deletions during sex chromosome evolution, dosage compensation evolves when essential or haplo-insufficient genes are deleted. In contrast, regulatory mechanisms may have evolved in lower vertebrates with the selective retention of haplo-insufficient genes [46].

A correct model of sex chromosome evolution is important for future work. For example, a large non-recombining SDR is likely predictive of a large inversion. Although allelic variations are the predominant means through which MSD genes are acquired in lower vertebrates, chromosomal inversions are a common strategy for the acquisition of MSD genes among vertebrates. However, the identification of inversion junctions from tens of millions of base pairs is still difficult. Future focus should be given to structural analysis around the borders of the SDR, between X and Y or W and Z sex chromosomes, and to epigenetic modification and the spatial architecture, which may regulate not only the acquisition of MSD genes but also the evolution of sex chromosomes. Additional efforts with amphibians and reptiles, as well as many taxa of teleost fish, will compensate for the knowledge gaps to enable a full understanding of the diversity of MSD genes and their chromosome evolution.

Gene knockout is well accepted for the functional analysis of genes. However, with the sex trait, a pure reliance on gene knockout could lead to incorrect conclusions. This is because the knockout of MSD genes is expected to cause sex reversal, but sex reversal can be achieved with many genes involved in the sex determination pathway (Table 1), not only with the MSD gene under study. For example, the knockout of the bcar1 gene in channel catfish led to the sex reversal of genetic males to neofemales [124], but bcar1 was later demonstrated not to be the MSD gene for channel catfish [87,125]. Sex reversal could be achieved readily in channel catfish when treated with estradiol, but the involved genes were very different from those involved in sex determination [131]. Similarly, with a Japanese strain of Nile tilapia (*Oreochromis niloticus*), knockouts of the maleness genes amhy, gsdf, or dmrt1, or the femaleness genes foxl2 or cyp19a1a, all led to sex reversal [132,133,134]. Thus, caution must be exercised when working with certain candidate MSD genes, especially with transcriptional factors and TGF-β cytokines. Fst mapping, when coupled with demonstrated expression profiles at critical time points in sex differentiation, should provide strong positional and expression evidence for the determination of MSD genes. When the size of the SDR is very large, there can still be multiple differentially expressed genes (DEGs) in the mapped SDR. Large differences in DEGs within the SDR, even between closely related species such as channel catfish and blue catfish, are suggestive of distinct MSD genes [135]. The detailed analysis of their spatial and temporal expression, as well as their epigenetic regulation, is required for the identification of MSD genes.

## Figures and Tables

**Figure 2 ijms-26-03282-f002:**
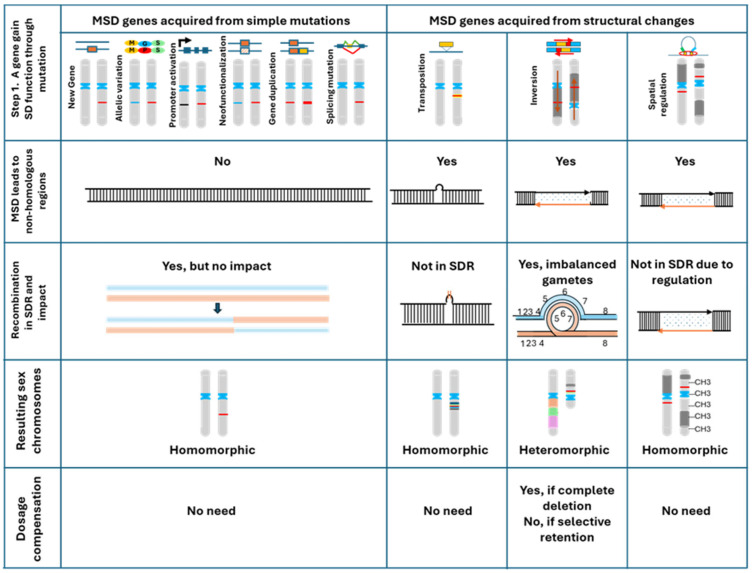
Schematic representation of the cause–effect model and predicted paths of sex chromosome evolution. The pathway by which the MSD genes were acquired (the cause) determines the ways in which the sex chromosomes evolve (the effect). The ways in which the MSD genes are acquired are categorized into two major categories: (1) simple mutations (left), such as gaining a new gene, base substitutions, mutations in the promoter region, neofunctionalization, subfunctionalization, tandem gene duplication, and splicing junction mutations, and (2) structural changes (right), such as large insertions, chromosomal inversion, and fusion and fission. A key element is whether non-homologous regions are created during MSD acquisition. With simple mutations, non-homologous regions are not involved; therefore, no recombination suppression occurs, leading to homomorphic sex chromosomes. Non-homologous sequences reduce recombination and, hence, involve recombination suppression, leading to heteromorphic sex chromosomes. In the case of major structural change, regions of non-homologous sequences are involved, e.g., in the inserted segment, which lead to recombination suppression. Although recombination suppression causes sequence degeneration and the accumulation of transposable elements, recurrent recombination in the reverted regions over time leads to deletions and rearrangements.

**Figure 3 ijms-26-03282-f003:**
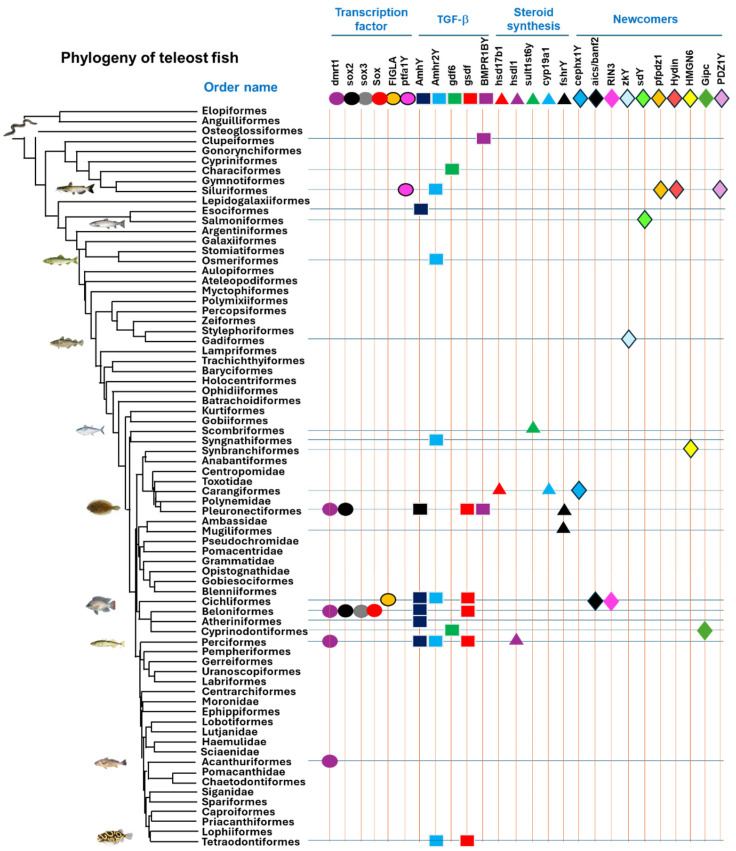
Divergent evolution of master sex determination (MSD) genes across teleost fish (Actinopterygii). Shown on the left is the phylogeny of teleost fish, adopted from Hughes et al. [96]. Shown on the right are MSD genes in various categories, such as transcription factors (circles), TGF-b (squares), genes involved in steroid synthesis (triangles), and “newcomers” (stars), illustrated in various colors and patterns, each representing a specific MSD gene. The symbols indicate where they each serve as MSD genes.

**Figure 4 ijms-26-03282-f004:**
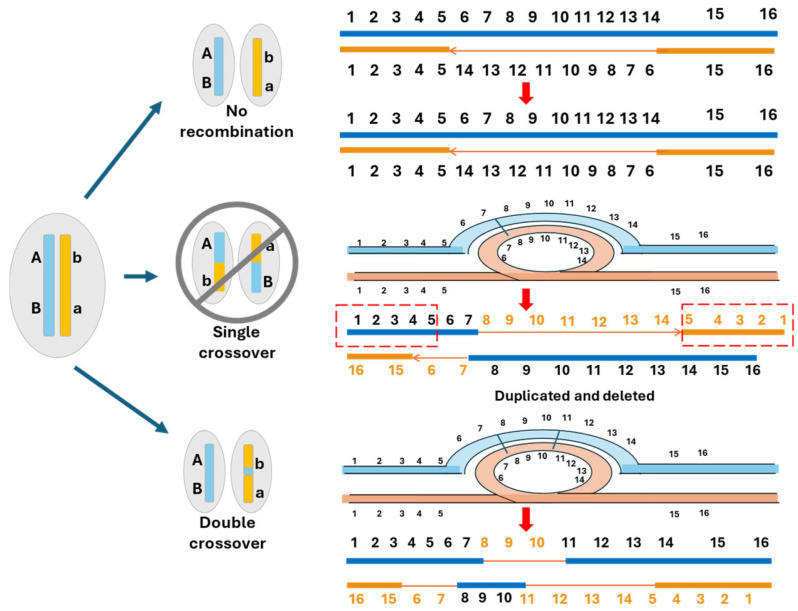
Schematic presentation of recombination involving chromosome inversion. A single crossover of the sex chromosome with a major inversion produces recombinant gametes that are not viable because of duplication and deletions. Double crossovers produce viable recombinant gametes but cause deletions and chromosomal rearrangements. Inversions, therefore, are the driving force of sex chromosome evolution.

**Table 1 ijms-26-03282-t001:** Examples of genes involved in male or female pathway of sex determination, which are antagonistic to each other.

Genes in the Male Pathway			Genes in the Female Pathway		
*SRY*	Sex-determining region on the Y chromosome	HMG-box transcription factor	*Foxl2*	Forkhead Box L2	Transcription factor
*sox9*	*SRY*-like, HMG-box-containing gene family, member 9	HMG-box transcription factor	*RUNX1*	RUNX Family Transcription Factor 1	Works with FOXL2 for fetal granulosa cell identity
*wt1*	Wilms’ Tumor 1	Transcription factor	*RSPO1*	R-spondin-1	Secreted intercellular signal
*dmrt1*	Doublesex and mab-3-related transcription factor 1	Transcription factor	*WNT4*	*Wnt* Family Member 4	Secreted intercellular signal
*sox3*	*SRY*-like, HMG-box-containing gene family, member 3	HMG-box transcription factor	*Cyp19a1*	Aromatase/estrogen synthetase	Converts androgens to estrogens
*Fgf9*	Fibroblast growth factor 9	Secreted intercellular signal	*Hsd17b1*	17β-Hydroxysteroid dehydrogenase 1	
*Amh & other TGF-β*	TGF-b superfamily	TGF signaling	β-catenin	Cell adhesion	Key player for Wnt pathway
			*MAP3K1*	Mitogen-activated protein kinase kinase kinase 1	Activates *Wnt4*/β-catenin/*FOXL2* pathway
*Newcomers*	*zkY*, *sdY*, *pfpdz1*, *hydin*, *cyce3*, *RIN3*, *FIGLA*, *znrf3*	To be defined	*Dax-1*	Dosage-sensitive sex reversal	Regulated by *Wnt4*
			Newcomers	*paics*, *banf2*	Unknown

**Table 2 ijms-26-03282-t002:** Diversity of sex determination (SD) genes and SD systems in vertebrates. The ZW sex system is highlighted in red. Question mark (?) indicates unknown master sex determination (MSD) genes, but their karyotypes are indicated. * indicates that inversion is likely involved because the whole chromosome is non-recombining, except a small PAR. The sign “-” indicates that functional validation was not conducted. The size of the sex determination region (SDR) is indicated for those that are well characterized. Ho stands for homomorphic, while Hetero stands for heteromorphic.

MSD Gene	Order or Major Group	Common Name	Species	MSD Gene Acquisition	Karyotype	Functional Validation	Sex System	Reference
Teleost fish								
Dmrt1	Beloniformes	Japanese medaka	*Oryzias latipes*	Allelic	Ho	Natural mutation	XY	[18]
Dmrt1	Beloniformes	Hainan medaka	*Oryzias curvinotus*	Allelic	Ho	-	XY	[19]
Dmrt1	Beloniformes	Northern medaka	*Oryzias Sakaizumii*	Allelic	Ho	-	XY	[20]
Dmrt1	Beloniformes	Hubbs’s medaka	*Oryzias hubbsi*	Inversion	Hetero	-	ZW	[21]
Dmrt1	Beloniformes	Javanese ricefish	*Oryzias javanicus*	Inversion	Hetero	-	ZW	[21]
Dmrt1	Perciformes	Spotted scat	*Scatophagus argus*	Allelic	Ho	-	XY	[22]
Dmrt1	Perciformes	Spotbanded scat	*Selenotoca multifasciata*	Allelic	Ho	-	XY	[23]
Dmrt1	Perciformes	Siamese fighting fish	*Betta splendens*	Allelic	Ho	-	XY	[24]
Dmrt1	Perciformes	Yellow drum	*Nibea albiflora*	Allelic	Ho 10 Mb SDR	-	XY	[25]
Dmrt1	Perciformes	Bighead croaker	*Collichthys lucidus*	Chromosome fusion generated Y (male 2n = 47; female 2n = 48)	Hetero	-	X_1_X_1_X_2_X_2_/X_1_X_2_Y	[26]
Dmrt1	Pleuronectiformes	Chinese tongue sole	*Cynoglossus semilaevis*	* Whole chromosome non-recombining but a small PAR	Hetero W larger	knockout	ZW	[27]
Dmrt1	Pleuronectiformes	Genko tongue sole	*Cynoglossus interruptus*	Allelic	Ho	-	ZW	[28]
Dmrt1	Acanthuriformes	Yellow croaker	*Larimichthys crocea*	Allelic	Ho	-	XY	[29]
sox2	Pleuronectiformes	Turbot	*Scophthalmus maximus*	Allelic	Ho	-	ZW	[30]
sox3Y	Beloniformes	Dwarf medaka	*Oryzias minutillus*	Allelic	Ho	Transgenic, knockout	XY	[31]
sox3Y	Beloniformes	Marmorated ricefish	*Oryzias marmoratus*	Allelic	Ho	Transgenic, knockout	XY	[31]
sox3Y	Beloniformes	Yellow finned medaka	*Oryzias profundicola*	Allelic	Ho	Transgenic, knockout	XY	[31]
sox3Y	Beloniformes	Indian ricefish	*Oryzias dancena*	Allelic	Ho	Transgenic, knockout	XY	[31]
sox7	Beloniformes	Celebes ricefish	*Oryzias celebensis*	Allelic	Ho	-	XY	[20]
sox7	Beloniformes	Matano ricefish	*Oryzias matanensis*	Allelic	Ho	-	XY	[20]
sox7	Beloniformes	Wolasi ricefish	*Oryzias wolasi*	Allelic	Ho	-	XY	[20]
sox7	Beloniformes	Daisy’s ricefish	*Oryzias woworae*	Allelic	Ho	-	XY	[20]
FIGLA	Cichliformes	Tilapia	*Oreochromis niloticus LG1*	Allelic	Ho	-	XY	[32]
ptfa1	Siluiriformes	Chinese longsnout catfish	*Leiocassis longirostris*	Allelic	Ho	-	XY	[33]
amhby	Esociformes	Norther pike	*Esox Lucius*	Tandem duplication	Ho	-	XY	[34,35]
amhby	Esociformes	Southern pike	*E. cisalpinus*	Tandem duplication	Ho	-	XY	[34]
amhby	Esociformes	Amur Pike	*E. reichertii*	Tandem duplication	Ho	-	XY	[34]
amhby	Esociformes	Muskellunge	*E. masquinongy*	Tandem duplication	Ho	-	XY	[34]
amhby	Esociformes	Chain pickerel	*E. niger*	Tandem duplication	Ho	-	XY	[34]
amhby	Esociformes	Olympic mudminnow	*Novumbra hubbsi*	Tandem duplication	Ho	-	XY	[34]
amhby	*Pleuronectiformes*	Olive flounder	*Paralichthys olivaceus*	Duplication and transposition	Ho	-	XY	[36]
amhY	Atheriniformes	Patagonian pejerrey	*Odontesthes hatcheri*	Duplication and transposition	Ho	Knockdown	XY	[37]
amhY	Atheriniformes	Silversides	*Odontesthes argentinensis*	Duplication and transposition	Ho	-	XY	[38]
amhY	Atheriniformes	Silversides	*Odontesthes nigricans*	Duplication and transposition	Ho	-	XY	[38]
amhY	Atheriniformes	Silversides	*Odontesthes piquava*	Duplication and transposition	Ho	-	XY	[38]
amhY	Atheriniformes	Silversides	*Odontesthes incisa*	Duplication and transposition	Ho	-	XY	[38]
amhY	Atheriniformes	Silversides	*Odontesthes smitti*	Duplication and transposition	Ho	-	XY	[38]
amhY	Atheriniformes	Silversides	*Odontesthes humensis*	Duplication and transposition	Ho	-	XY	[38]
amhY	Atheriniformes	Silversides	*Odontesthes regia*	Duplication and transposition	Ho	-	XY	[38]
amhY	Atheriniformes	Silversides	*Odontesthes mauleanum*	Duplication and transposition	Ho	-	XY	[38]
amhY	Atheriniformes	Silversides	*Odontesthes perugiae*	Duplication and transposition	Ho	-	XY	[38]
amhY	Atheriniformes	Pejerrey	*Odontesthes bonariensis*	Duplication and transposition	Ho		XY	[39]
amhY	Atheriniformes	Cobaltcap silverside	*Hypoatherina tsurugae*	Truncated duplication on Y	Ho	-	XY	[40]
amhY	Cichliformes	Nile tilapia	*Oreochromis niloticus*	Tandem duplication	Ho	-	XY Chr23	[41,42,43]
amhY	Beloniformes	Sulawesian meedaka	*Oryzias eversi*	Allelic	Ho	-		[44]
amhY	Perciformes	Black rockfish	*Sebastes schlegelii*	Duplication and transposition	Ho	Overexpression	XY	[45]
amhY	Perciformes	Korean rockfish	*Sebastes koreanus*	Duplication and transposition	Ho	-	XY	[45]
amhY	Perciformes	Rockfish	*Sebastes pachycephalus*	Duplication and transposition	Ho	-	XY	[45]
amhY	Perciformes	Threespine stickleback	*Gasterosteus aculeatus*	Inversion	Hetero	-	XY	[46]
amhY	Perciformes	Japan Sea stickleback	*Gasterosteus nipponicus*	Inversion followed by chromosome fusion	Hetero	-	X1X2Y	[47]
amhY	Perciformes	Blackspotted stickleback	*Gasterosteus wheatlandi*	Inversion followed by chromosome fusion	Hetero	-	X1X2Y	[47]
amhY	Perciformes	Brook stickleback	*Culaea inconstans*	Duplication on Y	Ho	-	XY	[48]
amhY	Perciformes	Common lumpfish	*Cyclopterus lumpus*	Duplication on Y	Ho	-	XY	[49]
amhY	Perciformes	Lingcod	*Ophiodon elongatus*	Duplication on Y	Ho	-	XY	[50]
amhY	Siluriformes	Southern catfish	*Silurus meridionalis*	Transposition	Ho 2.38 Mb SDR	-	XY	[51]
Amhr2Y	Siluriformes	Amur catfish	*Silurus asotus*	Transposition	Ho 400 Kb SDR	-	XY	[52]
*amhr2y*	Siluriformes	Lanzhou catfish	*Silurus lanzhouensis*	Transposition	Ho 400 Kb SDR	-	XY	[53]
Amhr2Y	Siluriformes	Pangasiidae catfishes	*Pangasianodon hypophthalmus*	Transposition	Ho 320 Kb SDR	-	XY	[54]
Amhr2Y	Siluriformes	Pangasiidae catfishes	*Pangasius djambal*	Transposition	Ho 320 Kb SDR	-	XY	[54]
Amhr2Y	Siluriformes	Pangasiidae catfishes	*Pangasianodon gigas*	Transposition	Ho 320 Kb SDR	-	XY	[54]
Amhr2Y	Siluriformes	Pangasiidae catfishes	*Pangasius bocourti*	Transposition	Ho 320 Kb SDR	-	XY	[54]
Amhr2Y	Siluriformes	Pangasiidae catfishes	*Pangasius conchophilus*	Transposition	Ho 320 Kb SDR	-	XY	[54]
Amhr2Y	Siluriformes	Pangasiidae catfishes	*Pangasius elongatus*	Transposition	Ho 320 Kb SDR	-	XY	[54]
Amhr2Y	Siluriformes	Pangasiidae catfishes	*Pangasius siamensis*	Transposition	Ho 320 Kb SDR	-	XY	[54]
Amhr2Y	Siluriformes	Pangasiidae catfishes	*Pangasius macronema*	Transposition	Ho 320 Kb SDR	-	XY	[54]
Amhr2Y	Siluriformes	Pangasiidae catfishes	*Pangasius larnaudii*	Transposition	Ho 320 Kb SDR	-	XY	[54]
Amhr2Y	Siluriformes	Pangasiidae catfishes	*Pangasius mekongensis*	Transposition	Ho 320 Kb SDR	-	XY	[54]
Amhr2Y	Siluriformes	Pangasiidae catfishes	*Pangasius krempfi*	Transposition	Ho 320 Kb SDR	-	XY	[54]
Amhr2Y	Siluriformes	Pangasiidae catfishes	*Pangasius sanitwongsei*	Transposition	Ho 320 Kb SDR	-	XY	[54]
amhr2Y	Osmeriformes	Ayu	*Plecoglossus altivelis*	Duplication and translocation	Ho 2.03 Mb SDR	Knockout	XY	[55]
amhr2Y	Syngnathiformes	Common seadragon	*Phyllopteryx taeniolatus*	Truncated duplication and transposition	Ho	-	XY	[56]
amhr2Y	Syngnathiformes	Alligator pipefish	*Syngnathoides biaculeatus*	Truncated duplication and transposition	Ho	-	XY	[56]
amhr2Y	Cichliformes	Midas Cichlids	*Amphilophus amarillo*	Duplication and transposition	Ho	-	XY	[57]
amhr2Y	Cichliformes	Midas Cichlids	*Amphilophus astorquii*	Duplication and transposition	Ho	-	XY	[57]
amhr2Y	Cichliformes	Midas Cichlids	*Amphilophus chancbo*	Duplication and transposition	Ho	-	XY	[57]
amhr2Y	Cichliformes	Midas Cichlids	*Amphilophus citrinellus*	Duplication and transposition	Ho	-	XY	[57]
amhr2Y	Cichliformes	Midas Cichlids	*Amphilophus flaveolus*	Duplication and transposition	Ho	-	XY	[57]
amhr2Y	Cichliformes	Midas Cichlids	*Amphilophus globosus*	Duplication and transposition	Ho	-	XY	[57]
amhr2Y	Cichliformes	Midas Cichlids	*Amphilophus labiatus*	Duplication and transposition	Ho	-	XY	[57]
amhr2Y	Cichliformes	Midas Cichlids	*Amphilophus sagittae*	Duplication and transposition	Ho	-	XY	[57]
amhr2Y	Cichliformes	Midas Cichlids	*Amphilophus tolteca*	Duplication and transposition	Ho	-	XY	[57]
amhr2Y	Cichliformes	Midas Cichlids	*Amphilophus viridis*	Duplication and transposition	Ho	-	XY	[57]
amhr2Y	Cichliformes	Midas Cichlids	*Amphilophus xiloaensis*	Duplication and transposition	Ho	-	XY	[57]
amhr2Y	Cichliformes	Midas Cichlids	*Amphilophus zaliosus*	Duplication and transposition	Ho	-	XY	[57]
amhr2Y	Perciformes	Yellow perch	*Perca flavescens*	Allelic	Ho	-	XY	[58]
amhr2Y	Perciformes	Balkhash perch	*Perca schrenkii*	Allelic	Ho	-	XY	[59]
amhr2Y	Perciformes	Walleye	*Sander vitreus*	Allelic	Ho	-	XY	[59]
amhr2Y	Tetraodontiformes	Pufferfish	*Takifugu rubripes*	Allelic	Ho	-	XY	[60]
amhr2Y	Tetraodontiformes	Pufferfish	*Takifugu obscurus*	Allelic	Ho	-	XY	[61]
amhr2Y	Tetraodontiformes	Pufferfish	*Takifugu ocellatus*	Allelic	Ho	-	XY	[61]
amhr2Y	Tetraodontiformes	Pufferfish	*Takifugu xanthopterus*	Allelic	Ho	-	XY	[61]
amhr2Y	Tetraodontiformes	Pufferfish	*Takifugu stictonotus*	Allelic	Ho	-	XY	[61]
amhr2Y	Tetraodontiformes	Pufferfish	*Takifugu porphyreus*	Allelic	Ho	-	XY	[61]
amhr2Y	Tetraodontiformes	Pufferfish	*Takifugu poecilonotus*	Allelic	Ho	-	XY	[61]
amhr2Y	Tetraodontiformes	Pufferfish	*Takifugu chrysops*	Allelic	Ho	-	XY	[61]
amhr2Y	Tetraodontiformes	Pufferfish	*Takifugu pardalis*	Allelic	Ho	-	XY	[61]
amhr2Y	Characiformes	Blind cave fish	*Astyanax mexicanus*	B chromosomes	B chr	Knockout	B chr	[62]
gdf6bB	Cyprinodontiformes	African killifish	*Nothobranchius furzeri*	Allelic	-	Knockout	XY	[63]
gdf6bY	Beloniformes	Philippine Medaka	*Oryzias luzonensis*	Allelic	Ho	Transgenic	XY	[64]
gsdfY	Cichliformes	Tilapia	*Oreochromis niloticus*	Allelic	Ho	Transgenic	XY	[65]
gsdfY	Perciformes	Sablefish	*Anoplopoma fimbria*	Allelic	Ho	-	XY	[66]
gsdfY	Pleuronectiformes	Atlantic halibut	*Hippoglossus hippoglossus*	Allelic	Ho 11.6 Mb SDR	-	XY Chr12	[67]
gsdfY	Tetraodontiformes	Pufferfish	*Takifugu niphobles*	Transposition	Ho		XY	[61]
gsdfY	Tetraodontiformes	Pufferfish	*Takifugu snyderi*	Transposition	Ho		XY	[61]
gsdfY	Tetraodontiformes	Pufferfish	*Takifugu vermicularis*	Transposition	Ho		XY	[61]
bmpr1ba	Pleuronectiformes	Pacific halibut	*Hippoglossus stenolepis*	Inversion compared to Chr9 of Atlantic halibut	Ho 12 Mb SDR	-	ZW Chr9	[68]
bmpr1bbY	Clupeiformes	Atlantic herring	* Clupea harengus *	Allelic	Ho	-	XY Chr8	[69]
fshrY	Mugiliformes	Flathead grey mullet	*Mugil cephalus*	Allelic	Ho	-	XY	[70]
fshrY	Pleuronectiformes	Senegalese sole	*Solea senegalensis*	Allelic	Ho	-	XY	[71]
Hsd17b1	Carangiformes	Yellowtail amberjack	*Seriola lalandi*	Allelic	Ho	-	ZW	[72]
Hsd17b1	Carangiformes	Greater amberjack	*Seriola dumerili*	Allelic	Ho	-	ZW	[72]
Hsd17b1	Carangiformes	Japanese yellowtail	*Seriola quinqueradiata*	Allelic	Ho	-	ZW	[72]
Hsd17b1	Carangiformes	California yellowtail	*Seriola dorsalis*	Allelic	Ho	-	ZW	[73]
Hsd17b1	Carangiformes	Oyster pompano	*Trachinotus anak*	Allelic	Ho	-	ZW	[74]
Cyp19a1	Carangiformes	Silver trevally	*Pseudocaranx georgianus*	Allelic	Ho	-	XY	[75]
Hsdl1 or *Tbc1d32*	Perciformes	European perch	*Perca fluviatilis*	Allelic	Ho	-	XY	[59]
sult1st6y	Scombriformes	Southern bluefin tuna	*Thunnus maccoyii*	Allelic	Ho	-	XY	[76]
sult1st6y	Scombriformes	Pacific bluefin tuna	*Thunnus orientalis*	Allelic	Ho	-	XY	[76]
cephx1Y	Carangiformes	Cobia	*Rachycentron canadum*	Allelic	Ho 4.04 Mb SDR	-	XY	[77]
Paics/banf2W	Cichliformes	Blue tilapia	*Oreochromis aureus LG3*	Duplication	Ho	-	ZW	[78,79]
Paics/banf2W	Cichliformes	Tanganyika tilapia	*Oreochromis tanganicae*	Duplication	Ho	-	ZW	[78,79]
Paics/banf2W	Cichliformes	Wami tilapia	*Oreochromis hornorum*	Duplication	Ho	-	ZW	[78,79]
Paics/banf2W	Cichliformes	Spotted tilapia	*Pelmatolapia mariae*	Duplication	Ho	-	ZW	[78,79]
RIN3	Cichliformes	-	*Chromidotilapia guntheri*	Allelic coding region	Ho	-	XY	[80]
zkY	Gadiformes	Atlantic cod	*Gadus morhua*	Allelic	Ho	-	XY	[81]
zkY	Gadiformes	Arctic cod	*Arctogadus glacialis*	Allelic	Ho	-	XY	[81]
zkY	Gadiformes	Pacific cod	*Gadus macrocephalus*	Allelic	Ho	-	XY	[81]
sdY	Salmoniformes	Salmonids	*13 species* *Salmonids*	Transposition/translocation	-	Transgenic and knockout	XY	[82,83]
pfpdz1	Siluiriformes	Yellow catfish	*Pelteobagrus fulvidraco*	Chromatin architecture, epigenetic regulation	Ho	Overexpression, knockout	XY	[84,85]
Hydin	Siluiriformes	Channel catfish	*Ictalurus punctatus*	Epigenetic regulation	Ho 8.9 Mb SDR	Methylation blocker	XY	[86,87]
? (newcomer)	Siluiriformes	Ussuri catfish	*Pseudobagrus ussuriensis*	Epigenetic regulation	Ho 16.83 Mb SDR	-	XY	[88]
HMGN6/CYCE3	Synbranchiformes	Zig-zag eel	*Mastacembelus armatus*	Allelic	Ho 7.0 Mb SDR	-	XY	[89]
Gipc PDZ1Y?	Cyprinodontiformes	Eastern mosquitofish	*Gambusia holbrooki*	-	Ho	-	XY	[90]
?	Cyprinodontiformes	Western mosquitofish	*Gambusia affinis*	W chr larger, fusion?	Hetero	-	ZW	[90]
Mammals								
Sry	Mammals	All mammals		Inversion *	Hetero	Knockout, overexpression	XY	[11]
Birds								
Dmrt1	Birds	All birds		Inversion *	Hetero	Allelic knockout	ZW	[12]
Reptiles								
?		Python and boa snakes	*-*	-	Hetero	-	ZW	[13]
?		Amazonian puffing snakes and viper snakes	*-*	-	Ho	-	XY	[13]
Amphibians								
DM-W		African clawed frog	*Xenopus laevis*	Allelic	Ho	Transgenic	ZW	[91]
Bod1l		European green toad	*Bufo viridis*	Allelic	Ho		XY	[14]

## Data Availability

Not applicable.

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
