# Peer review of "The Cause–Effect Model of Master Sex Determination Gene Acquisition and the Evolution of Sex Chromosomes"

_ijms, 2025, doi:10.3390/ijms26073282_

Round 1
Reviewer 1 Report
Comments and Suggestions for Authors
This paper discusses how the acquisition of master sex determination (MSD) genes influences the evolution of sex chromosomes in vertebrates. While the classical model suggests that sex chromosomes inevitably degrade over time, many lower vertebrates, especially teleost fish, retain homomorphic sex chromosomes. By analyzing over 30 identified MSD genes, the paper reveals that different mechanisms of MSD gene acquisition—such as simple mutations, gene duplications, or chromosomal rearrangements—lead to varying degrees of recombination suppression and sex chromosome differentiation. The proposed cause-effect model suggests that the mode of MSD gene acquisition directly determines whether sex chromosomes remain homomorphic or become heteromorphic. The paper highlights that a critical factor in sex chromosome evolution is whether non-homologous regions are formed during MSD gene acquisition. Chromosomal inversions, for example, create sequences that initially suppress recombination but can still undergo recombination over time, contributing to degradation and, in some cases, dosage compensation mechanisms.
This paper provides a unifying framework to explain why some vertebrates retain homomorphic sex chromosomes while others develop highly differentiated ones. The cause-effect model offers a novel perspective on how different mechanisms of MSD gene acquisition influence sex chromosome evolution, helping to reconcile discrepancies observed across vertebrate species. The findings have broad implications for understanding sex determination systems and chromosome evolution in fish and other vertebrates. This manuscript is well written and insightful, only a few small issues (see comments in detail) need to be addressed before its acceptance for publication.
Comments in detail
- Figure 1. Enlarge fonts
- Define “DM-W”
- Table 2. Clarify the meaning of “-.” Does it indicate missing information?
- L163-165. Please verify the accuracy of the claim: “Although both XY and ZW sex determination systems exist in reptiles, no master sex-determining (MSD) genes have been identified.” It appears that several key sex-determining genes, such as dmrt1 and amh, have been identified in reptiles, reflecting the diversity of sex determination mechanisms in this group. Additionally, some reptiles, like the bearded dragon, exhibit interactions between genetic sex determination (GSD) and temperature-dependent sex determination (TSD), highlighting the evolutionary flexibility of reptilian sex determination systems.
- The claim that the "FIGLA-like gene is the MSD gene for Nile tilapia" may not be entirely correct. In Nile tilapia (Oreochromis niloticus), sex determination is influenced by both genetic and environmental factors. The primary sex-determining gene is amh (anti-Müllerian hormone), located on the sex chromosome (LG23), specifically the amh-Y duplication (amhy). Males typically follow an XX-XY system, where amhy on the Y chromosome promotes testis development. The FIGLA-like gene on LG1 may also play a role in sex determination, but it is unlikely to be the master sex-determining gene (MSD).
- Suggest enlarging fonts on left side.
- L333-372. It may be helpful to discuss the factors that lead to structural changes in chromosomes, which can result in the formation of sex chromosomes. These changes, such as chromosomal inversions, occur due to errors in homologous recombination, DNA breakage, and repair. Contributing factors include misaligned meiotic crossing-over, double-strand breaks (caused by radiation, chemicals, or oxidative stress), transposable elements, and non-homologous end joining. Additionally, replication errors, chromosomal instability disorders, viral integration, and unequal sister chromatid exchange can also play a role. These mechanisms lead to either paracentric inversions (which exclude the centromere) or pericentric inversions (which include the centromere), ultimately influencing gene expression, evolution, and genetic disorders.
- Please update the reference. This paper was published, see Nature Reviews Genetics volume 26, pages 59–74 (2025).
Author Response
Reviewer 1
This paper discusses how the acquisition of master sex determination (MSD) genes influences the evolution of sex chromosomes in vertebrates. While the classical model suggests that sex chromosomes inevitably degrade over time, many lower vertebrates, especially teleost fish, retain homomorphic sex chromosomes. By analyzing over 30 identified MSD genes, the paper reveals that different mechanisms of MSD gene acquisition—such as simple mutations, gene duplications, or chromosomal rearrangements—lead to varying degrees of recombination suppression and sex chromosome differentiation. The proposed cause-effect model suggests that the mode of MSD gene acquisition directly determines whether sex chromosomes remain homomorphic or become heteromorphic. The paper highlights that a critical factor in sex chromosome evolution is whether non-homologous regions are formed during MSD gene acquisition. Chromosomal inversions, for example, create sequences that initially suppress recombination but can still undergo recombination over time, contributing to degradation and, in some cases, dosage compensation mechanisms.
This paper provides a unifying framework to explain why some vertebrates retain homomorphic sex chromosomes while others develop highly differentiated ones. The cause-effect model offers a novel perspective on how different mechanisms of MSD gene acquisition influence sex chromosome evolution, helping to reconcile discrepancies observed across vertebrate species. The findings have broad implications for understanding sex determination systems and chromosome evolution in fish and other vertebrates. This manuscript is well written and insightful, only a few small issues (see comments in detail) need to be addressed before its acceptance for publication.
Thank you for your positive review of our work.
- Figure 1. Enlarge fonts
The fonts have been increased.
- Define “DM-W”
Revised, as suggested
- Table 2. Clarify the meaning of “-.” Does it indicate missing information?
“-” indicates functional validation was not conducted, and this is now added in Table 2 headings.
- L163-165. Please verify the accuracy of the claim: “Although both XY and ZW sex determination systems exist in reptiles, no master sex-determining (MSD) genes have been identified.” It appears that several key sex-determining genes, such as dmrt1 and amh, have been identified in reptiles, reflecting the diversity of sex determination mechanisms in this group. Additionally, some reptiles, like the bearded dragon, exhibit interactions between genetic sex determination (GSD) and temperature-dependent sex determination (TSD), highlighting the evolutionary flexibility of reptilian sex determination systems.
Thanks for this comment. Yes, these genes have been implicated, but not confirmed, as MSD genes. We have incorporated this into the text.
- The claim that the "FIGLA-like gene is the MSD gene for Nile tilapia" may not be entirely correct. In Nile tilapia (Oreochromis niloticus), sex determination is influenced by both genetic and environmental factors. The primary sex-determining gene is amh (anti-Müllerian hormone), located on the sex chromosome (LG23), specifically the amh-Y duplication (amhy). Males typically follow an XX-XY system, where amhy on the Y chromosome promotes testis development. The FIGLA-like gene on LG1 may also play a role in sex determination, but it is unlikely to be the master sex-determining gene (MSD).
Thank you. This has been changed now to: FIGLA-like gene was reported to be the key sex regulator of LG1 sex determination in tilapia (Curzon et al.).
- Suggest enlarging fonts on left side.
Thanks. The fonts are now increased.
- L333-372. It may be helpful to discuss the factors that lead to structural changes in chromosomes, which can result in the formation of sex chromosomes. These changes, such as chromosomal inversions, occur due to errors in homologous recombination, DNA breakage, and repair. Contributing factors include misaligned meiotic crossing-over, double-strand breaks (caused by radiation, chemicals, or oxidative stress), transposable elements, and non-homologous end joining. Additionally, replication errors, chromosomal instability disorders, viral integration, and unequal sister chromatid exchange can also play a role. These mechanisms lead to either paracentric inversions (which exclude the centromere) or pericentric inversions (which include the centromere), ultimately influencing gene expression, evolution, and genetic disorders.
Thank you! These are very great comments. We have now incorporated this paragraph under Conclusion and Perspectives.
- Please update the reference. This paper was published, see Nature Reviews Genetics volume 26, pages 59–74 (2025).
The reference is now updated

Reviewer 2 Report
Comments and Suggestions for Authors
The canonical model of vertebrate sex chromosome evolution predicts one way of trend toward degradation. However, most sex chromosomes in lower vertebrates are homomorphic. The study give the new theory that cause-effect model of master sex determination gene acquisition and evolution of sex chromosome. It is important for understanding mechanism of sex determination in fish. The manuscript may be accepted after minor revise.
- Pls add order name in figure 3 for easy understanding evolution fo MSD genes.
- Pls add the mechanism "selective retention of haploinsufficient genes is apparently an alternative strategy" in sbstract.
Author Response
Reviewer 2:
The canonical model of vertebrate sex chromosome evolution predicts one way of trend toward degradation. However, most sex chromosomes in lower vertebrates are homomorphic. The study give the new theory that cause-effect model of master sex determination gene acquisition and evolution of sex chromosome. It is important for understanding mechanism of sex determination in fish. The manuscript may be accepted after minor revise.
Thank you for your nice review of our work.
- Pls add order name in figure 3 for easy understanding evolution for MSD genes.
Thank you. Added now
- Pls add the mechanism "selective retention of haploinsufficient genes is apparently an alternative strategy" in abstract.
Thank you. Added now